# The Effects of Paddy Cultivation and Microbiota Members on Arsenic Accumulation in Rice Grain

**DOI:** 10.3390/foods12112155

**Published:** 2023-05-26

**Authors:** Esra Ersoy Omeroglu, Asli Bayer, Mert Sudagidan, Veli Cengiz Ozalp, Ihsan Yasa

**Affiliations:** 1Basic and Industrial Microbiology Section, Biology Department, Faculty of Science, Ege University, Bornova, 35100 Izmir, Türkiye; 91190000729@ogrenci.ege.edu.tr (A.B.); ihsan.yasa@ege.edu.tr (I.Y.); 2Department of Medical Biology, Medical School, Atilim University, 06830 Ankara, Türkiye; msudagidan@gmail.com (M.S.); cengiz.ozalp@atilim.edu.tr (V.C.O.)

**Keywords:** rice grain, arsenic, methane, metabarcoding, qRT-PCR, safe food

## Abstract

Access to safe food is one of the most important issues. In this context, rice plays a prominent role. Because high levels of arsenic in rice grain are a potential concern for human health, in this study, we determined the amounts of arsenic in water and soil used in the rice development stage, changes in the *arsC* and *mcrA* genes using qRT-PCR, and the abundance and diversity (with metabarcoding) of the dominant microbiota. When the rice grain and husk samples were evaluated in terms of arsenic accumulation, the highest values (1.62 ppm) were obtained from areas where groundwater was used as irrigation water, whereas the lowest values (0.21 ppm) occurred in samples from the stream. It was observed that the abundance of the Comamonadaceae family and *Limnohabitans* genus members was at the highest level in groundwater during grain formation. As rice development progressed, arsenic accumulated in the roots, shoots, and rice grain. Although the highest *arsC* values were reached in the field where groundwater was used, methane production increased in areas where surface water sources were used. In order to provide arsenic-free rice consumption, the preferred soil, water source, microbiota members, rice type, and anthropogenic inputs for use on agricultural land should be evaluated rigorously.

## 1. Introduction

Agricultural use has a 37% share of the land area at the global level. Most of this land is used to produce rice, a staple food that act as an important energy (70%) and protein (50%) source for more than half of the world’s population [1,2,3,4,5]. However, prevalent arsenic (As) pollution in rice soil and the high uptake of As by plants pose health risks [6]. As, a geogenetic pollutant, is widely distributed throughout the world. Indeed, with the contamination of drinking water and cultivated plants, As joins the food chain and has posed a significant threat to the health of approximately 150 million people worldwide in recent years [3,7,8,9,10]. Rice, which is one of the basic foodstuffs, accumulates a significant amount of As. It has also been reported that rice accumulates almost 10 times more As than grain products such as corn, wheat, and barley [5,9,10,11,12]. In this context, the effects of As exposure have been studied; as a result, As has been found to be associated with acute and chronic human health problems such as cancer, skin lesions, diabetes, and cardiovascular disease [9,10].

Therefore, examining the microorganism-mediated biotransformation of As in rice and its production areas is important because this plays a critical role regarding the fate and toxicity of As [3,13]. According to the results obtained from studies on *Escherichia coli*, *Pseudomonas aeruginosa*, and *Staphylococcus* sp., the resistance developed against As compounds is regulated by an operon known as the *ars* operon, which consists of at least three genes: *arsB*, *arsC*, and *arsR* [14,15,16,17,18]. The *arsC* gene is involved in As detoxification by encoding a cytoplasmic arsenate reductase that plays a role in the reduction of arsenate (As(V)) to arsenite (As(III)) [18,19,20].

Although rice production plays a critical role in sustaining food security on the global scale, another concern is the fact that rice production accounts for a significant part of global methane (CH_4_) emissions and has an effect on global climate change [21]. At this point, studies to be carried out on the methyl coenzyme M reductase gene *mcrA* [22], the gene required for CH_4_ production, constitute one of the critical steps in determining the necessary methodology for detection and prevention in agricultural areas.

Different methods are used to measure gene expression. However, real-time quantitative polymerase chain reaction (qRT-PCR), which is also used in this study, is a fast, highly sensitive, highly accurate, and easy quantitative enumeration method that enables the real-time amplification of target genes [18,23,24,25,26].

In this study, regarding rice production areas and the products consumed as foods in Balıkesir Gönen, Türkiye, it was hypothesized that the determination of the total As quantity at different stages during the cultivation of rice varieties could contribute to the implementation of necessary precautions in rice production. Furthermore, by determining the As dynamics at the *arsC* functional gene level, a reduction in the amount of As that can accumulate in rice grains can be achieved. It has also been hypothesized that, with the detection of the *mcrA* gene region, it is possible to reduce the amount of CH_4_ gas (one of the greenhouse gases) from such agricultural areas, which will contribute to the reduction in the effect of greenhouse gases on the earth. For these purposes, qRT-PCR was used as a current, fast, and reliable technique.

At the same time, in order to obtain current information from rice fields at various stages of production, the sampling of water, soil, and different plant parts was carried out, and analyses were performed to determine the total As content of the samples as well as the P, K, Mn, Fe, Cu, and Zn contents. Thus, this study aimed to contribute to sustainable agriculture by taking measures to prevent the negative effects of agricultural activities on the ecological balance and, as a result, to prevent adverse effects on the surrounding ecosystem.

In this study, we focused on determining the copy number of *arsC* by qRT-PCR during the cultivation of rice and the variability in the microbial community by the metagenomic method. While making the general evaluation, the As content of the samples and the copy number of *mcrA* were evaluated. The ultimate goal was to obtain data that can be used to determine the production conditions that can be created to reduce the amount of As accumulating in the grain in order to ensure safe rice consumption. As well as the effect of the microbial remediation processes on As accumulation in rice, the phytoremediation feature of rice has a significant effect [27]. Therefore, rice development stages and microbial cycles should be evaluated together to ensure safe rice production and consumption.

## 2. Materials and Methods

### 2.1. Sampling Site Selection and Sample Collection

The Balıkesir-Gönen region (Türkiye), where paddy production is intense and the Turkish paddy variety Gönen rice is cultivated, was chosen as the sampling area. In total, lands from three different regions were assessed. The physical and chemical properties of the water used for irrigation and agricultural soil were taken into account to assess the land. For this purpose, light, medium, and heavy soil criteria were taken into account as soil properties. In the agricultural areas in this region, areas that irrigate with dam water (Gönen Yenice Dam), stream water (Gönen Stream), and groundwater were also added to the selection criteria. In this context, the Danakumu (irrigation with light soil and groundwater), Ulukır (irrigation with medium soil and the Gönen Stream), and Gündoğan (irrigation with heavy soil and dam water) regions were used as sampling areas.

At the first stage of sampling, soil and irrigation water sampling was carried out before the sowing of seeds from the paddy fields. Soil and water sampling continued at the critical stages of rice development. Root and shoot samples were collected during the paddy development stages from the first shoot formation to grain formation (see Appendix A). At the last stage, grain and husk samples were added to all samples. During sampling, the sampling point coordinates and temperature were determined.

### 2.2. Arsenic Analysis of Various Plant Tissues, Soil, and Water

The total As concentrations in the root, shoot, grain, and husk samples obtained at different stages of rice cultivation were determined. At the same time, seven metals (including As) were analyzed in the soil and water samples obtained as a result of the sampling carried out before and after seed sowing and after the first sprouting period, the first tillering period, the grain formation period, and the completion of the harvest. They were analyzed for their total As concentrations using inductively coupled plasma mass spectrometry (ICP-MS) [7]. All heavy metal analyses were performed using the EPA Method 3051 analysis method [28,29]. For chemical analyses, the multi element ICP QC standard solution (QCS-27) (27E) was used. This solution contains 27 elements in 2 to 5% HNO_3_ + traces of HF (QCS-27).

### 2.3. Preparation of Total Bacterial DNA

Total bacterial DNA isolations were performed in order to carry out molecular studies at the functional gene level and also to carry out metagenomic studies from soil and water samples obtained within the scope of sampling. For isolation of the total genomic DNA from the water samples, 2 L of water was filtered using the required amount of membrane filter. Since the filter (pore diameter 0.22 µm) was larger than approximately 15 cm in diameter, the filter was cut into many small pieces under aseptic conditions, and then the kit procedure was started. Soil sampling was conducted to represent the whole sample, and DNA was isolated in 10 different tubes for the same sample. Finally, the total DNA was obtained. At this stage, Norgen Biotek’s Genomic DNA Isolation Kit (News Medical, Cat. 24700, Thorold, ON, Canada), Water RNA/DNA Purification Kit (Product #26480, Thorold, ON, Canada), and Soil DNA Isolation Plus Kit (Product # 64000, Thorold, ON, Canada) were used as the DNA isolation kits. For DNA integrity control, gel electrophoresis with 1% agarose was performed, and spectrophotometric measurements were completed [29]. At the same time, purity controls were made by considering the ratio of A_260_/A_280_ of the total DNA obtained by using nanodrop.

### 2.4. Primers

The *arsC* gene, which is responsible for the first step of As(V) biotransformation by encoding the arsenate reductase enzyme, and the *mcrA* gene, which is the methyl coenzyme M reductase gene required for CH_4_ production, were selected as the target gene regions. The aim was to determine the gene copy numbers of these gene regions by using the qRT-PCR technique. The total bacteria content was also detected using designed primer probes. For this purpose, the primers indicated in Table 1 were used (Table 1).

### 2.5. Quantitative Real-Time PCR (qRT-PCR) Conditions

After the DNA was extracted, ligation, transformation, and plasmid extraction were performed. Ampicillin was added to the medium as a selectivity factor for the selection of resistant colonies at the cloning stage [30]. *Escherichia coli* JM109 was used as a potent cell in the transformation stage [31]. Bacterial growth in the SOC medium was evaluated for transformation control [32].

Plasmid concentrations required for standard formation were prepared, and standard graphic studies were completed. To form the standard curve, serial dilutions containing 10-fold dilutions were prepared from each sample, and 3 parallels were used for each dilution. At this stage, studies were carried out so that the PCR efficiency was 95% and the coefficient value was greater than 0.98. qRT-PCR studies were completed using Fast Start Syber Green dye for each dilution [33]. The qRT-PCR reaction mix was prepared by adding 5 µL of template DNA, 1.6 µL of MgCl_2_, 1 µL of forward primer, 1 µL of reverse primer, 2 µL of Syber Green enzyme, and ultrapure water up to a 20 µL total reaction volume. The numbers of arsenate-reducing bacteria and CH_4_-producing bacteria in the samples were then calculated using the created standard using the equation “2.5/arsenate-reducing (or CH_4_ producing) gene copy = an arsenate-reducing or CH_4_ producing cell”.

PCR amplification was carried out in 1–50 ng of template DNA and 20 µL of the prepared reaction mixture. The qRT-PCR conditions were as follows: a single cycle of 10 min at 95 °C, followed by 40 cycles at 95 °C for 10 s, 55–65 °C for 5 s, 72 °C for 15–25 s, and then a single cycle of 30 s at 40 °C. In the melting curve analysis, the temperature was increased from 60 °C to 95 °C by increasing the temperature by 0.1 °C in each cycle. The trials were completed with a waiting period of 8 s between cycles [22,34].

### 2.6. Next-Generation Sequencing (NGS) and Metabarcoding

For the identification of bacterial communities, the NGS method was used. Preparation of the16S metabarcoding library was carried out according to the manufacturer’s instructions, as described in the Part # 15044223 Rev. B document (Illumina, Inc., San Diego, CA, USA). The forward and reverse primers (1 µM) included overhang adapter sequences (forward primer 5′-TCGTCGGCAGCGTCAGATGTGTATAAGAGACAGCCTACGGGNGGCWGCAG-3′ and reverse primer 5′-GTCTCGTGGGCTCGGAGATGTGTATAAGAGACAGGACTACHVGGGTATCTAATCC-3′) that were used to amplify the V3 and V4 regions of the prokaryotic 16S rRNA regions with 2 × KAPA HiFi HotStart Ready Mix (KK2602, 07958935001, Roche, Germany) to give a total amplicon PCR mixture volume of 25 µL. The obtained forward sequences were analyzed with the operational taxonomic unit (OTU) approach using SILVA NGS 1.4 software (reference version 138.1) and SINA v1.2.10 for ARB SVN (revision 21008) with BLASTn 2.2.30+ at the phylum and family-levels. The Shannon diversity index was also determined with SILVA NGS software (version 28) [35].

## 3. Results

### 3.1. Characteristics of the Sampling Sites and Samples

Within the scope of the study, sampling was carried out at five different stages based on the paddy development stages. During sampling, temperature measurements were conducted, and specific codes were given to the paddy samples. Temperatures from 19 °C to 29 °C were recorded, and a total of 54 samples were obtained. Information about the sampling points and sample codes is shown in Table 2.

### 3.2. Chemical Analysis

When the chemical analysis results of the soil samples collected before seed sowing were evaluated, it was seen that soil samples collected from the Gündoğan region had the highest values in terms of all evaluated parameters except for phosphorus concentration. When the values of three different sampling regions were examined in terms of the phosphorus concentration, it was seen that there was an opposing order, where the highest value (73.436 ppm) occurred in the Ulukır region and the lowest value (45.65 ppm) occurred in the Gündoğan region. The highest As content was found in the soil of the Gündoğan region, with a value of 32.6 ppm (Figure 1A; see Appendix A).

Regarding the amounts of As found in the water samples, it was seen that the amount of As increased as the rice farming stages progressed (see Appendix A). In the first sampling period, there was no significant amount of As in the water collected at all sampling points, whereas the As value in the water samples was 11.987 ppm in the water from the Ulukır region collected in June. The potassium content was the highest in all water samples (Figure 1B; see Appendix A).

For the sludge samples, the highest As value occurred in the Gündoğan samples. It was seen that the highest As value (20.474 ppm) was observed in the sludge samples collected during the first rice tillering period. In terms of the other parameters, a similar picture was observed in the sludge samples from the three sampling regions (Figure 1C; see Appendix A).

In the rice root samples, the highest P (779.368 ppm) value occurred in the Gündoğan root sample (GDR1) at the first germination stage. The highest K (2509.63 ppm) value was found in the Danakumu root sample (DKR2) at the first tillering stage. The Ulukır root sample (UKR1), which was in the first germination stage, had the highest Cu (7.224 ppm) content. The highest Mn and Zn values occurred in the samples collected from the Gündoğan (GDR3) (182.18 ppm) and Danakumu (DKR3) (54.42 ppm) regions, where grain formation had begun. When the roots containing the highest Fe (5811.64 ppm) and As (41.683 ppm) contents were examined, it was seen that the Ulukır samples (UKR2) were in the first tillering stage (Figure 1D; see Appendix A).

In the Danakumu and Gündoğan root samples, the amount of accumulated As increased as the paddy development stages progressed. However, a sudden increase in the amount of As in the root samples occurred in the first tillering stage. As expected, the amount of P in the root samples showed a serious decrease with the onset of grain formation. A similar situation occurred for the amount of K in the roots. On the contrary, in the Mn and Zn samples, the amount in the roots increased with the first grain formation (Figure 1D; see Appendix A).

In general, the amount of As accumulated in the shoot increased with the formation of the first grain. The highest value (4.7 ppm) was found in the shoot samples from the Danakumu region (see Appendix A). The shoots from the Ulukır region had the highest potential in terms of the Fe (1403.084 ppm), Cu (7.083 ppm), and Zn (26.758 ppm) concentrations in the rice shoots when the first germination period started. With the formation of the first grain, it was seen that the amount of Mn in the shoots from all regions increased to the highest level. The highest P (1391.29078 ppm) and K (10096.011 ppm) values in the shoots occurred in the samples collected from the Gündoğan region at the time of the first germination. In all shoot samples, the amount of As accumulated in the shoot decreased in the first tillering period but increased significantly with grain formation (Figure 1E; see Appendix A).

The highest As value (0.41 ppm) in the rice grains occurred in the sample (DKP1) obtained from the Danakumu region, where irrigation was conducted with groundwater. In the husk samples, the highest value (1.62 ppm) occurred in the sample (DKK1) from the same region. This value was also the highest As value in grain. The lowest As content (0.21 ppm) in both grain and husk samples occurred in the sample (UKP1) obtained from the Ulukır region irrigated by the Gönen Stream (Figure 1F; see Appendix A).

When the obtained grains were examined without separation, the highest As value (0.48 ppm) occurred in samples from the Gündoğan region. In all samples the lowest As content occurred in the rice samples cultivated in the Ulukır region (see Appendix A). It was seen that Gündoğan husk samples ranked first in terms of their K, Mn, and Fe contents. However, when rice grains were compared in terms of their P contents, it was seen that the Danakumu region ranked first with a value of 3125.3 ppm (Figure 1F; see Appendix A).

### 3.3. Quantification of the Total Bacteria and arsC and mcrA Genes in Paddy Samples by qRT-PCR

Regarding the analysis of the water samples with the qRT-PCR technique, the total number of bacteria gradually increased in the water samples collected from the Danakumu region. There was a decrease in the samples from the Ulukır region at first, but the values increased considerably in the last sampling period. In fact, the highest bacterial load (8.3 × 10^6^ copies reaction^−1^) in the water samples occurred in the UKS4 sample. In the water samples collected from the Gündoğan region, there was a sudden decrease in the second sampling period, but values stabilized afterwards (Figure 2).

The highest bacterial count (2.3 × 10^6^ concentration reaction^−1^) was found in the soil samples from the Danakumu region. In the sludge samples from this region, although the value was very high in the first sampling period (1.0 × 10^8^ concentration reaction^−1^), it decreased in the following periods (4.4 × 10^6^ concentration reaction^−1^). However, in the sludge samples, the highest value occurred in the samples obtained from this region (DKB1). In the Ulukır region, after a small increase in the beginning, a decrease occurred in the final sampling period. During rice development in the Gündoğan region, there was an increase in the number of bacterial microbiota members in the soil (Figure 2).

When all sludge samples were evaluated, the highest *arsC* gene copy number values were found in the Danakumu sludge samples (DKB1, DKB2, DKB3, and DKB4). However, within the scope of all *arsC* gene copy number results, the highest value (1.1 × 10^5^ concentration reaction^−1^) occurred in the sludge sample (UKB1) obtained from the Ulukır region in the first sampling period. When water samples were evaluated, it was seen that the lowest *arsC* value occurred in the Gündoğan samples. On the other hand, the region with the highest *arsC* gene copy number in the soil samples was Gündoğan. Although the *arsC* gene copy number decreased in the sludge obtained from the Danakumu and Ulukır regions in the second sampling period, it increased in the Gündoğan sludge samples (Figure 3A,B).

The highest value in terms of the *mcrA* gene copy number occurred in the sludge sample that was first collected from the Danakumu region. However, the gene copy number values decreased as the rice development stages progressed. The lowest *mcrA* gene copy number values for the sludge samples were obtained from the Ulukır region. Just as in the sludge samples, the highest *mcrA* gene copy number occurred in the water sample obtained from Danakumu in the first sampling period. Although the value decreased over time in the Danakumu water samples, the number of *mcrA* gene copies increased gradually in the Ulukır and Gündoğan water samples. In the soil samples, the highest value occurred in samples collected from the Danakumu region (Figure 3A,B).

### 3.4. Bacterial Diversity of Water, Soil, and Sludge Samples from Rice Farms

As a result of the metagenomic analysis (see Appendix A), it was seen that the members of the Proteobacteria phylum dominated in terms of both species diversity and species abundance. This situation was exhibited at the highest level, especially in the DKS2, DKS3, and GDS1 samples. Following the Proteobacteria phylum, the highest abundance and diversity ratios were found for the Bacteriodota, Verrucomicrobiota, and Actinobacteriota phyla. In particular, the abundance rate of Verrucomicrobiota members was higher in the GDS3 sample than in other samples. Nitrispirota members showed a different picture in terms of the species abundance in only one sample (GDB4) (Figure 4).

When the samples were examined in terms of phylum diversity, samples of groundwater from the Danakumu region (DKS1) and dam water from the Gündoğan region (GDS1) had the lowest diversity levels. These samples contained members of the Bacteriodota and Proteobacteria phyla as a common feature. Members of the Proteobacteria and Bacteriodota phyla were found in all samples, whereas Nanoarchaeota members were only found in the water sample (DKS3) collected from the Danakumu region during the first tillering period (Figure 4).

Although the Gammaproteobacteria and Bacteroidia members were found in all samples collected from the areas where rice cultivation was carried out, the greatest species diversity and abundance for the Gammaproteobacteria members occurred in the DKS3 and GDS1 samples. On the other hand, Alphaproteobacteria members were found to be biota members in all samples except for the GDS1 sample. The highest number of class members occurred in the GDB3 sample collected from the Gündoğan region at the time of first tillering (see Appendix A).

In terms of the order of microbiota members, all samples contained Burkholderiales members as a common feature. In terms of the species abundance and diversity, the DKS3 and DKS4 samples collected from the Danakumu region ranked first. At the same time, unlike the other samples, the DKS3 sample had Omnirtophales members. A similar situation was observed for the GDB4 sample collected from the Gündoğan region. Spirochaetales and Methylococcales members were only detected in the GDB4 sample. Ricketsiales members only existed in the GDS3 sample. Rhodobacterales members were only present in the DKS2 and DKS4 samples, and their species abundance was richest in the DKS2 sample. Acidobacteriales members were only present in the GDB3 sample. Bdellovibrionales members were only found in the DKS3 sample (see Appendix A).

For water samples from the Gündoğan region, the ordo diversity was higher in the GDS3 sample, which was collected at the first tillering stage in the summer period. A similar situation occurred for water samples collected from the Danakumu region. Although Pseudomonadales members were dominant in terms of the species diversity and abundance in the first water samples collected from the Gündoğan region, this situation changed in water samples collected during the later stages of rice development (see Appendix A).

Comamonadaceae family members were found in all samples except for three sludge samples (UKB1, UKB2, and UKB3). The highest abundance and diversity of these family members occurred in the DKS3 and DKS4 samples collected from the Danakumu region. Pseudomonadaceae members were mostly found in the first water sample (GDS1) collected from the Gündoğan region. In this sample, Flavobacteriaceae members ranked first in terms of abundance. Some family members were found to exist in only one sample each (Figure 5).

In the first water sample (DGS1) collected from the Gündoğan region, members of the *Pseudomonas* genus were the most abundant. At the same time, species from the *Massilia* genus, which are bacteria that support plant growth by dissolving phosphorus in the soil, were only present in this sample. Members from the *Flavobacterium* genus were only found in six water samples (DKS1, DKS2, GDS1, GDS3, GDS4, and UKS1). The species abundance and diversity were the most intense in GDS1. Members of the *Limnohabitans* genus were found in 11 samples (GDS1, GDS2, GDS3, GDS4, UKS1, UKS2, UKS3, UKS4, DKS2, DKS3, and DKS4); among these, members of the *Limnohabitans* genus were found more frequently in the DKS2, DKS3, and DKS4 samples than in other samples. However, members of the *Flavobacterium* genus were found to be concentrated in the GDS1 sample collected from the Gündoğan region (Figure 6, see Appendix A).

## 4. Discussion

As poisoning of ground and surface waters is a current problem faced on earth [36]. With the increases in global warming and anthropogenic pollution rates, the rate of As poisoning is increasing significantly [37]. However, intensive agricultural practices are carried out with As-contaminated water sources, since the basic nutritional elements needed by the exponentially increasing human population are insufficient. As accumulation occurs in agricultural products obtained as a result of using As-contaminated water sources for irrigation [38]. As a result of the consumption of many agricultural products by groups living at different stages in the food chain, As exposure of the environment and all living groups occurs. Therefore, when we evaluate As in terms of the “one health” approach [39], studies conducted to determine and eliminate As pollution rates in such habitats are of importance. In this context, bioremediation of As from soil samples is particularly noteworthy as a green technology [40]. Such studies are very important; however, based on the principle that “one kilogram of pollution removal is equivalent to one gram of pollution prevention”, determination of the increase in As pollution and the variability of metabolizing bacterial species in this context is also necessary. However, one of the important factors in As pollution is the emission of CH_4_, one of the greenhouse gases that supports the increase in global warming [41], which should also be monitored during agricultural processes. In this context, rice, which is one of the most widely grown and consumed foodstuffs worldwide, was used as the main material in the study. At the same time, the high accumulation of As in rice and the fact that rice fields account for approximately 40% of global CH_4_ emissions were the most important criteria in the selection of the study area. The total CH_4_ emissions originating from rice cultivation in 10 provinces in which 94.8% of the rice harvest takes place in Türkiye has increased by 21.4% in the last 10 years. Balıkesir-Gönen, which was the sampling region in this study, is the location for 85% of Türkiye’s rice production.

In this context, within this study, As pollution and As accumulation in rice were determined in different regions during rice cultivation. At the same time, quantitative analyses of bacteria with the gene regions necessary for the metabolism of As (the *arsC* gene region) and methane production (the *mcrA* gene region) during the rice development stages were conducted by qRT-PCR. Metagenomic analyses were carried out in the same samples. In this way, important steps can be taken to prevent the accumulation of As in rice by ensuring that the necessary green technology measures are taken during rice farming, where As accumulation and methane production are intense. At the same time, it is thought that this information will contribute to the development of measures to reduce gas emissions in paddy agricultural lands that add to global warming in terms of CH_4_ gas production at a high rate. Therefore, we have the opportunity to positively modify the paddy, As, and methane equation, which we can describe as the “gorgeous trio”.

In this context, when all data were evaluated to ensure that the targeted recommendations were presented, the region with the highest arsenic content before the addition of rice and the water source to the soil was the Gündoğan region (GDS1, Figure 1A). In terms of water resources used in agricultural lands, the Gönen Stream, which is used for the irrigation of the Ulukır region, has the highest As content, and the As content increases as rice development progresses (UKS1, UKS2, UKS3, and UKS4, Figure 1B). Thus, it was observed that the lowest As content in terms of the three water sources occurred in the groundwater used for the irrigation of the Danakumu region (DKS1, DKS2, DKS3, and DKS4, Figure 1B). However, the highest As value occurred in the soil sample (GDS1) collected as a result of the saturation of the paddy fields with water in the first stage, again in the Gündoğan region, which is irrigated by the Yenice Dam (Figure 1A, see Appendix A). As a general result, with the first germination period, the amount of As in the sludge samples of all regions showed a significant increase. This diversity in the As concentration is due to differences in the soil in which rice cultivation occurs and the sources used as irrigation water (Figure 1C, see Appendix A). In particular, the processes controlling the transfer of aquifers to various water resources and thus the increase in the amount of As in water resources are not fully understood. However, it can be said that the main source of As is related to iron hydroxide in the aquifer [42]. When evaluated from this point of view, the iron content was the highest in soil (GDT1), water (GDS1), and sludge (GDB3) samples from the Gündoğan region. In general, the greatest As accumulation was observed in this region before rice cultivation started. When only the situation in the water resources was evaluated, the highest As (6515–11.99 ppm) content was found in the samples (UKS1, UKS2, UKS3, and UKS4) collected from the Gönen Stream, which is used for irrigation of the Ulukır region. In the water samples collected in June, the highest value occurred and the negative effects of increases in anthropogenic pollution and hot air on As contamination in the Gönen Stream were revealed (Figure 1).

With root formation, the highest As values occurred in the samples from the Ulukır region, which contained the lowest As concentration in the sludge samples but the highest concentration in the irrigation water. However, this value occurred in the first tillering period, and an approximately five-fold decrease in the amount of As in the root occurred with grain formation. In other samples, the amount of As accumulated in the root during grain formation increased (Figure 1, see Appendix A). As is abundant in rice and As fertilizers, herbicides, and pesticides. As minerals in the soil pass into the water. Later, rice absorbs As, and then rice is cooked and As is passed on to people. The use of rice straw as animal feed causes As to pass to animals as secondary consumers and then to humans again. When the conversion of As into volatile forms (arsine [AsH**_3_**], monomethylarsine [CH**_3_**AsH**_2_**], dimethylarsine [(CH**_3_**)**_2_**AsH], and trimethylarsine [(CH**_3_**)**_3_**As]) is included, the arsenic cycle in rice agriculture emerges [43]. For this reason, As accumulation may occur at different rates from root to grain during the paddy cultivation stages. One of the determining factors in accumulation rates is the As and iron contents in the agricultural land. With the growth of plants, the presence of higher iron concentrations in the root soil during the reproductive phase causes the formation of iron plaques on the root surface, which trap As and prevent its uptake by plants. As a result, the co-precipitation of As with iron released from crystallized iron plaques results in the loosening of iron plaques from the root surface. Thus, the soil As concentration increases in the final stage of cultivation, which contributes to an increase in the concentration in plant parts [44]. In our study, the data showed this effect.

Although the amount of As accumulated in the shoots increased with grain formation, the highest values occurred in the shoots obtained from the Danakumu region (DKSH3), which is irrigated with groundwater. The lowest values were observed in samples from the Gündoğan Region, where irrigation is conducted with water from the Yenice dam (GDSH3). Although the Danakumu region ranks second in terms of As accumulation in sludge, the highest As accumulation in exile was observed in this region. On the contrary, the accumulation was at the lowest level in the rice shoots grown in the Gündoğan region (GDSH3), which showed the highest sludge (GDB3) As value. Therefore, irrigation with dam water appears to be the most appropriate method. When the rice grain and husk samples were evaluated in terms of As accumulation, the highest values occurred in samples from the Danakumu region (DKK1), whereas the lowest values occurred in samples from the Ulukır region (UKP1). When the samples were evaluated without separating the husk, the highest values occurred in samples from Gündoğan (GDG1) and the lowest values occurred in samples from the Ulukır region (UKG1) (Figure 1, see Appendix A). Thus, lands where groundwater is used as an irrigation source should be examined more carefully in the evaluation of the paddy parts consumed as foods to determine whether they are safe in terms of their As contents. Since groundwater is a closed system, aquatic habitats, such as dams and streams, do not have the ability to dilute toxic compounds. Thus, our results reveal that groundwater, for which pollution monitoring is very difficult, has removal and regeneration properties that are cause for serious alarm. Considering that global warming and anthropogenic pollution are gradually increasing [45], the increase in the demand for groundwater for agricultural land prevents the safe consumption of obtained food.

As-contaminated irrigation water greatly affects the rise of As in the soil and its subsequent accumulation in rice grains. As accumulation in rice is highly dependent on its bioavailability, even in As-rich soil, and is influenced by various factors such as the soil type; physicochemical parameters; presence of other elements; composition of minerals such as iron, phosphorus, sulfur, and the soil–rhizosphere–plant system; rhizospheric microorganisms and their activities; organic matter and related microbial populations; water regime; and nutritional status [46].

Although the amount of As in soil, water, and sludge is very important, the microbiota in the habitats where rice cultivation is carried out is of great importance. With the occurrence of biochemical transformations, there are variations in the amounts of different compounds. Therefore, the species abundance and diversity should be evaluated in order to make accurate assessments in such habitats [47].

As expected, the total number of bacteria in the Gönen Stream and groundwater increased remarkably with the warming of the air; on the contrary, there was a serious decrease in the total amount of bacteria in the Yenice dam water. Because the Danakumu region, which is irrigated with groundwater, contains the most microbiota members in terms of the soil total bacteria count, the highest bacterial count occurred in the Danakumu sludge sample (DKB1). Although the bacterial load in the underground water, which is the water source for the Danakumu region, gradually increased, there was a numerical decrease in the sludge samples in the following sampling periods. Although the bacterial load in the Yenice Dam water (Gündoğan region) gradually decreased, the number of bacteria contained in the sludge sample gradually increased (Figure 2). The highest *arsC* gene copy numbers were also found in the sludge samples, consistent with the occurrence of the highest bacterial load in the Danakumu DKB1 sample. At the same time, the highest *mcrA* gene copy number was found in this sample. With the onset of the first germination period, there was a decrease in the number of *arsC* gene copies in the sludge samples from the Danakumu and Ulukır regions; however, in contrast, there was an increase in the samples from the Gündoğan region. Whereas the *mcrA* gene copy number was high, consistent with the high total bacterial load in the first Danakumu sludge sample obtained by fully saturating the agricultural land with water before germination started, a decrease in the copy number of this gene region occurred with the onset of germination (Figure 3). Variations in anaerobic conditions are an important factor affecting the species diversity and abundance of dominant biota members. At the same time, changes in the physicochemical properties of habitats also cause changes in the distribution of microbiota members [47,48,49]. The decrease in these values also coincides with the fact that the groundwater that is used for irrigation of the Danakumu region had a lower *mcrA* gene copy number in the samples from the later stages of rice development.

There was no significant relationship between the amounts of As contained in soil, water, and sludge samples and the *arsC* gene copy number present in these samples. There are many gene regions that function within the scope of bacterial As metabolism [50]. As a result of the expression of each gene region, different products are released. As a result of the activity of the *arsC* gene region, arsenate is reduced [51] and it turns into arsenite, which is a more water-soluble form that can be absorbed by plants through their roots [52]. In this way, As accumulation occurs in different parts of the plants [44]. In this context, in order to ensure safe food consumption that does not threaten public health, the *arsC* gene region is intensively investigated in terms of As accumulation in food [52,53,54]. In this study, no significant relationship was found between the amount of As in sludge and the *arsC* gene copy number in the same samples. This can be explained by the fact that arsenic contamination did not occur long ago, and therefore, members of the microbiota have not had enough time to obtain the necessary gene regions. However, as a result of the highest *arsC* gene copy numbers occurring in the Danakumu sludge samples (Figure 3), the greatest As accumulation was observed in shoot, rice grain, and husk samples obtained from this region (Figure 1, see Appendix A). At this stage, it is thought that this situation is due to the ability of the paddy to accumulate As or because of the stabilization properties of As in the soil [27]. Although the total As content of Danakumu sludge samples ranks second, the microbiota members in the soil and groundwater coincide with the highest As accumulation found in rice grain and husk samples obtained as a result of paddy cultivation in terms of the species diversity and abundance. Soil naturally contains As, and as a result, some As is found in all foods. However, because rice is grown in wetlands, it has the capacity to absorb more As. Compared with other crops, such as wheat, rice can absorb up to 10 times more arsenic [11]. For this reason, it is necessary to determine all of the factors that support As accumulation in rice development, as they cause As exposure for all living groups that use rice products as foods. As a result, it is necessary to take various measures (fertilizers effective on nitrification or the use of a biofertilizer that can have a positive effect on the As cycle in the soil) to change this situation in paddy farming because people are exposed to high amounts of As by drinking arsenic-containing water, irrigating food crops with these waters, and consuming foods that have been exposed to As in industrial processes [38].

In order to prevent food consumption and exposure to toxic chemicals such as As, cultivation methods should be developed to reduce their accumulation. For this reason, in order to evaluate all physicochemical cycle dynamics, it is necessary to determine the dominant microbiota members in the aquaculture stages. When it comes to rice farming, microbial communities that provide methane production come into play, along with As pollution. One of the most important global methane gas sources is paddy fields. All global problems should be solved by approaching from the framework of a single health approach.

In this context, the evaluation should not only involve functional As resistance genes but also genes that contribute to an increase in the As concentration. For this reason, the stages in which changes in the emission of methane, which is the second most prominent greenhouse gas after carbon dioxide, occur during rice farming should be determined, and evaluations should be conducted with multiple approaches. Methanogens include phylogenetically diverse taxa belonging to the phylum Euryarchaeota. Many ecologically important methanogen families (Methanobacteriaceae, Methanomicrobiaceae, Methanosaetaceae, Methanosarcinaceae, and Methanocellaceae) have been identified in rice paddy soils. Methane can be metabolized anaerobically as well as aerobically. Aerobic methanotrophs are also found in the phyla Proteobacteria and Verrucomicrobia [55]. The result showed that the Proteobacteria and Bacteroidota members were common in all samples and that the Proteobacteria members ranked first in terms of the species abundance and diversity. These phyla were followed by members of Verrucomicrobiota (except DKS2 and GDS1), Planctomycetota (DKS1, DKS2 and UKS4), and Actinobacteriota. The sample (DKS3) collected from the Danakumu region in the summer during the first tillering period has the highest potential in terms of the abundance and diversity of Proteobacteria members and is irrigated with groundwater. It emerged as the only sample in which members of the symbiotic archaea group Nanoarchaeota were observed. Sludge samples are habitats with a more intense species diversity and abundance when compared with water and soil. In this context, the highest phylum diversity occurred in the sludge sample collected from the Gündoğan region during grain formation, with 21 phylum members (Figure 4).

When all of the results had been evaluated, it was revealed that the Danakumu region, where groundwater is used as an irrigation source, had the greatest As accumulation in rice grain and husk samples. During grain formation, the diversity and abundance of the members of the Comamonadaceae family (Figure 5), as well as the members of the *Limnohabitans* genus (Figure 6) (which generally live planktonically in freshwater lakes [56]), were the highest in the groundwater.

In the samples from the Danakumu region, it was revealed that as paddy development progressed, the amounts of As accumulated in the root and shoot also increased, resulting in the highest As accumulation in the husk and rice grains. These obtained data became more meaningful as the Danakumu samples contained the highest *arsC* gene copy numbers. The second highest level of accumulation occurred in rice from the Gündoğan region, which is irrigated with dam water. The same picture was observed in this region. Accumulation in roots and shoots gradually increased, resulting in an increase in the grain samples as a result. There was a gradual increase in the number of *mcrA* gene copies, the last component of the gorgeous trio, in the Ulukır and Gündoğan regions, where the surface water sources, the Gönen Stream and Yenice Dam, were used to obtain irrigation water. This is an expected result, as the air temperature increase and the effects of anthropogenic pollution inputs can be seen immediately.

## 5. Conclusions

People are exposed to high levels of As by drinking arsenic-contaminated water, using these water sources as sources of irrigation for food crops, and consuming arsenic-contaminated food as a result of industrial processes. Rice, which is one of the basic foodstuffs consumed all over the world, is cultivated under anaerobic conditions and exposed to high levels of As contamination in the wetlands where it is grown. For this reason, the current study was carried out to determine the changes in the amount of As accumulated in rice by assessing the functional gene pathway and to support the data with a metagenomic study. As a final word, the highest rate of As accumulation was found in the rice grains cultivated with groundwater, which is more difficult to pollute compared with surface water. Demand for groundwater is gradually increasing due to the decrease in surface water due to the increase in global warming and the effects of anthropogenic pollution, as the remaining areas are also unusable. However, due to the high cost of monitoring these water resources in terms of their capacity to threaten public health, the food obtained from these resources and agricultural areas is increasingly becoming a great risk. The factors that can be effective in As accumulation are the As content of the water and soil used in rice farming, the phytoremediation ability of rice, the stabilization properties of As on land, and the microbiota members present. However, it would be beneficial to include all microbial physiological cycles in the evaluation in order to make the most accurate interpretation of the final result. Many gene regions are required for the metabolism of As, and these gene regions can vary in different species. Therefore, it is necessary to examine the effects of microbiota members and their resistance genes on As accumulation in the environment and food in detail. Although researchers are working on different bacteria and genes responsible for arsenic uptake, transport, and/or detoxification in rice plants to produce more suitable crops for consumption, cultivation under various field conditions with different microbiota members and subsequent quality rice production remain unsolved problems. The data obtained in this study reveal the land and irrigation sources in the sampling area with high potential for arsenic risk. In rice agriculture, such studies should be carried out first and the soil and irrigation resources that contain the most suitable microbiota members in terms of arsenic bioremediation and pose a low arsenic risk should be selected. As a result, important steps will be taken to reach a safe environment and safe food production with biofertilizer or vaccine culture applications that contain the necessary microbiota members with the necessary gene regions.

## Figures and Tables

**Figure 1 foods-12-02155-f001:**
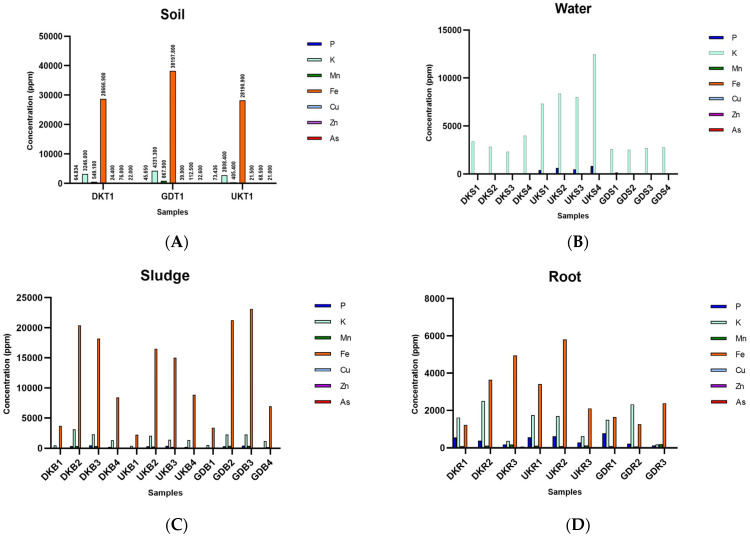
Chemical analysis of the paddy samples. (**A**) Soil, (**B**) water, (**C**) sludge, (**D**) root, (**E**) shoot, and (**F**) grain and husk (P: phosphorus, K: potassium, Mn: manganese, Fe: iron, Cu: cupper, Zn: zinc, As: arsenic. DK: Danakumu, UK: Ulukır, and GD: Gündoğan regions. T: soil, S: water, B: sludge, R: root, SH: shoot, G: grain and husk, K: husk, and P: grain samples).

**Figure 2 foods-12-02155-f002:**
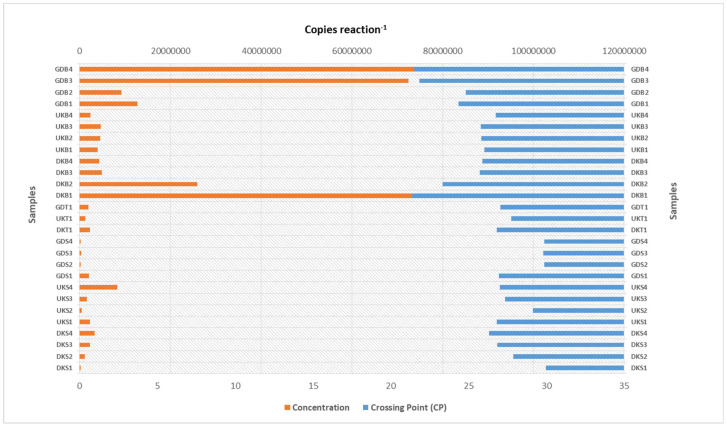
qRT-PCR amplification results related to the total bacteria count (DK: Danakumu, UK: Ulukır, and GD: Gündoğan regions. T: soil, S: water, B: sludge, R: root, SH: shoot, G: grain and husk, K: husk, and P: grain samples).

**Figure 3 foods-12-02155-f003:**
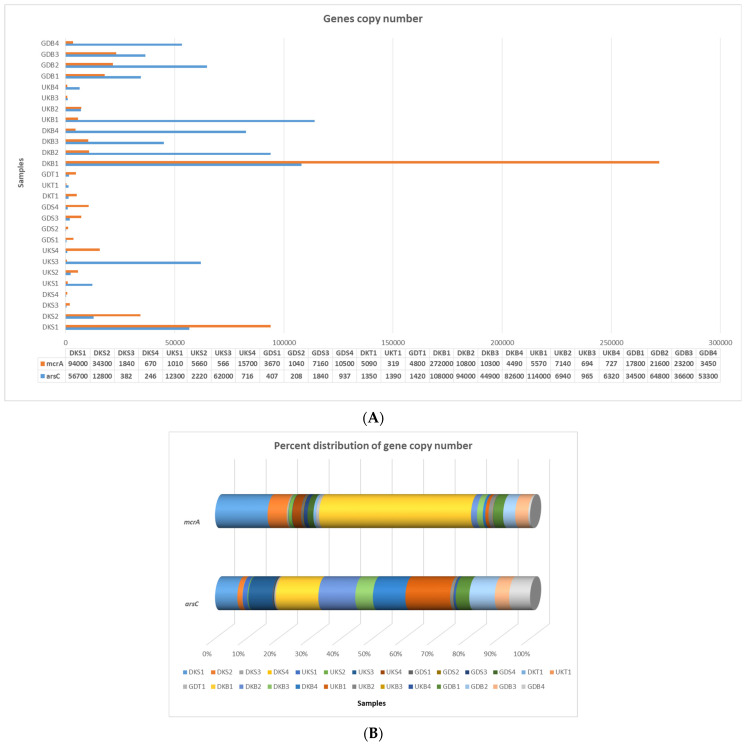
qRT-PCR amplification results for the *arsC* and *mcrA* gene regions. (**A**) Gene copy number, (**B**) percent distribution (DK: Danakumu, UK: Ulukır, and GD: Gündoğan regions. T: soil, S: water, B: sludge, R: root, SH: shoot, G: grain and husk, K: husk, and P: grain samples).

**Figure 4 foods-12-02155-f004:**
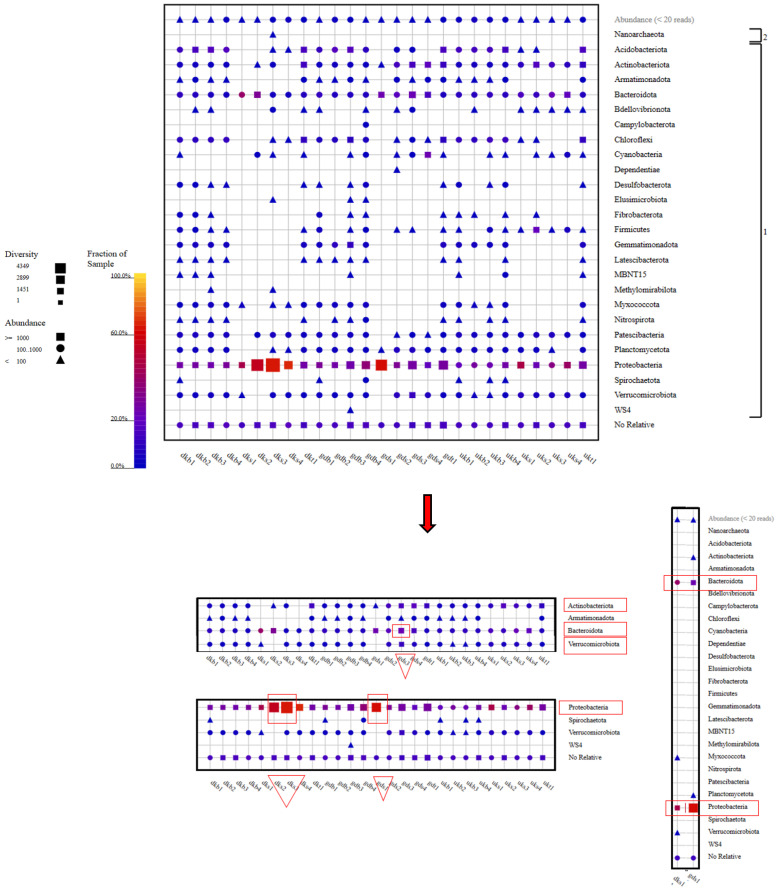
Phylum-level taxonomic fingerprint of water, soil, and sludge samples collected from rice farms. (The figure was created using SILVA NGS 1.4 v138.1) (DK: Danakumu, UK: Ulukır, and GD: Gündoğan regions. T: soil, S: water, B: sludge, R: root, SH: shoot, G: grain and husk, K: husk, and P: grain samples).

**Figure 5 foods-12-02155-f005:**
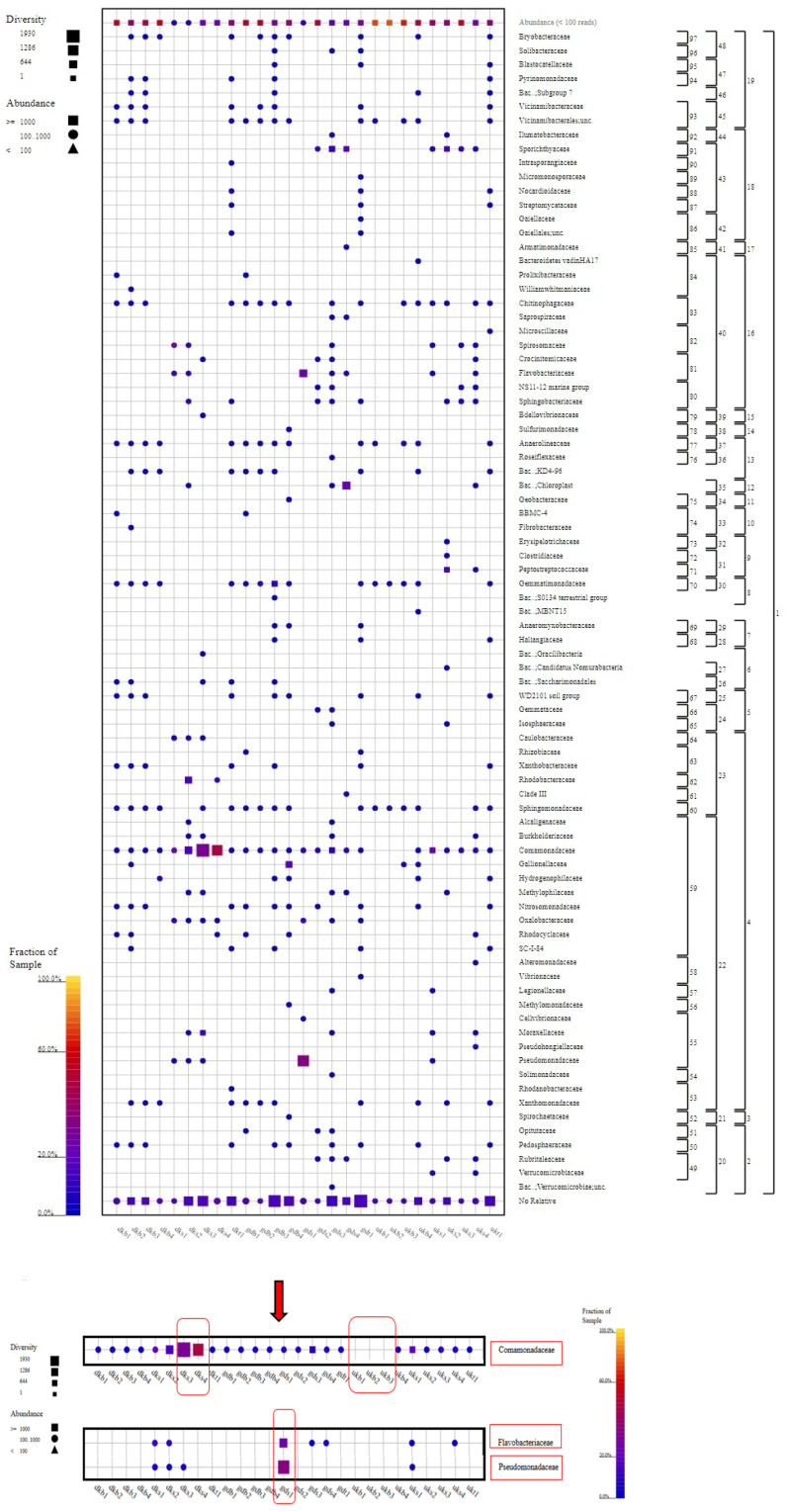
Family-level taxonomic fingerprint of water, soil, and sludge samples from rice farms (figure was created using SILVA NGS 1.4 v138.1). DK: Danakumu, UK: Ulukır, and GD: Gündoğan regions. T: soil, S: water, B: sludge, R: root, SH: shoot, G: grain and husk, K: husk, and P: grain samples).

**Figure 6 foods-12-02155-f006:**
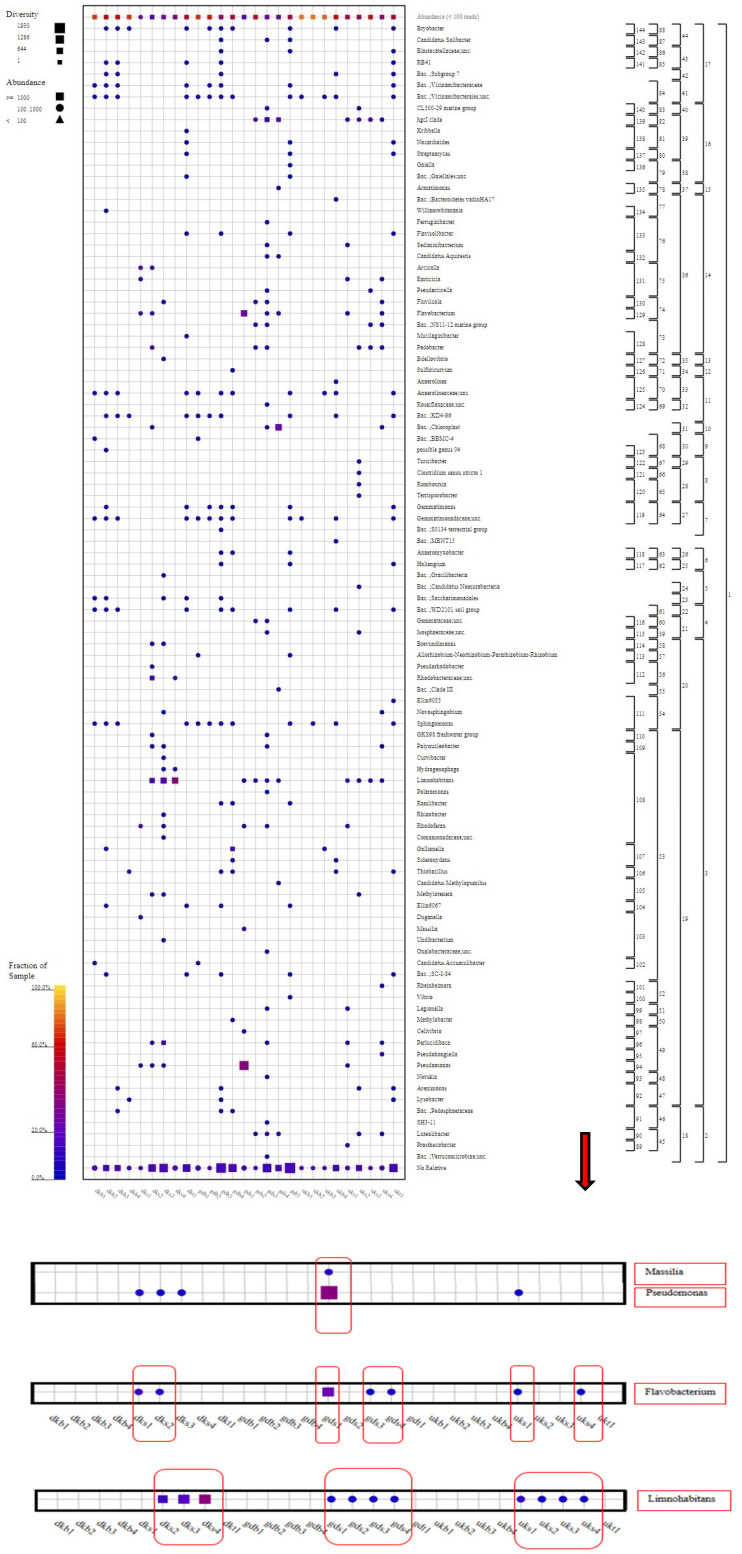
Genus-level taxonomic fingerprint of water, soil, and sludge samples from rice farms (figure was created using SILVA NGS 1.4 v138.1. DK: Danakumu, UK: Ulukır, and GD: Gündoğan regions. T: soil, S: water, B: sludge, R: root, SH: shoot, G: grain and husk, K: husk, and P: grain samples).

**Table 1 foods-12-02155-t001:** Characteristics of primers used in the qRT-PCR assays.

Primer Code		Primer Sequence (5′-3′)
Arsenate reductase	*arsC-F*	AATGAGATWCTGATATGAGCAACAT
	*arsC-F2*	TTGTAATGAGATWCTGATATGAGCA
	*arsC-R*	CCRCTGTTGCGGATCATYTC
	*arsC-R2*	CCCATATCBGCAATRAGTTT
	*arsC-PMGB*	FAM-ATGATCCGCAACAG-MGB-Q530
Methyl coenzyme M reductase	*Mcr_aF*	ATG BTS TAY GAC CAG WTV TGG
	*Mcr_aR*	TAYCCGAAGAAKCCSAGTC
	*Mcr_P1*	FAM-TAC ATG TCA GGT GGT GTM GGA TT-BHQ1
	*Mcr_P2*	FAM-TAC ATG AGC GGT GGT GTY GGT TT-BHQ1
	*Mcr_R1*	CACTTCGGTGGTTCCCAGA
	*Mcr_R2*	CACTTCGGTGGATCGCA
	*Mcr_R3*	CACTTCGGTGGATCTGTT
Total bacteria	*BAC338F*	ACTCCTACGGGAGGCAG
	*BAC805R*	GACTACCAGGGTATCTAATC
	*BAC516*	FAM-TGCCAGCAGCCGCGGTAATAC-BHQ1

**Table 2 foods-12-02155-t002:** Various characteristics of the sampling regions and samples selected from the paddy fields.

Region	Sample Code	Sample	Date	Time	Temperature	Depth	Irrigation Water	Soil Characteristic	Coordinates
Danakumu	DKS1	Water	6 May 2021	15:03	24 °C	**25 cm**	Ground water	Light	Lat → 40°08′30″ N Long → 027°39′30″ E
	DKT1	Soil							
	DKB1	Sludge	7 May 2021	14:30	29 °C				
	DKS2	Water	26 May 2021 (the first sprouding)	10:48	21 °C				
	DKB2	Sludge							
	DKR1	Root							
	DKSh1	Shoot							
	DKS3	Water	12 June 2021 (the first tillering)	09:43	23 °C				
	DKB3	Sludge							
	DKR2	Root							
	DKSh2	Shoot							
	DKS4	Water	17 September 2021 (grain formation)	14:13	29 °C				
	DKB4	Sludge							
	DKR3	Root							
	DKSh3	Shoot							
	DKP1	Grain							
	DKK1	Husk							
	DKG1	Grain + Husk							
Ulukır	UKS1	Water	6 May 2021	15:45	24 °C	**25 cm**	Gönen River	Medium	Lat → 40°15′33″ N Long → 027°36′44″ E
	UKT1	Soil							
	UKB1	Sludge	7 May 2021	15:36	29 °C				
	UKS2	Water	26 May 2021 (the first sprouding)	12:00	19 °C				
	UKB2	Sludge							
	UKR1	Root							
	UKSh1	Shoot							
	UKS3	Water	12 June 2021 (the first tillering)	11:15	23 °C				
	UKB3	Sludge							
	UKR2	Root							
	UKSh2	Shoot							
	UKS4	Water	17 September 2021 (grain formation)	15:05	27 °C				
	UKB4	Sludge							
	UKR3	Root							
	UKSh3	Shoot							
	UKP1	Grain							
	UKK1	Husk							
	UKG1	Grain + Husk							
Gündoğan	GDS1	Water	6 May 2021	17:00	24 °C	**25 cm**	Gönen Yenice Dam	Heavy	Lat → 40°10′48″ N Long → 027°37′54″ E
	GDT1	Soil							
	GDB1	Sludge	7 May 2021	15:09	29 °C				
	GDS2	Water	26 May 2021 (the first sprouding)	11:32	20 °C				
	GDB2	Sludge							
	GDR1	Root							
	GDSh1	Shoot							
	GDS3	Water	12 June 2021 (the first tillering)	10:11	23 °C				
	GDB3	Sludge							
	GDR2	Root							
	GDSh2	Shoot							
	GDS4	Water	17 September 2021 (grain formation)	14:40	29 °C				
	GDB4	Sludge							
	GDR3	Root							
	GDSh3	Shoot							
	GDP1	Grain							
	GDK1	Husk							
	GDG1	Grain + Husk							

## Data Availability

Data is contained within the article or Appendix A.

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
