# Peer review of "The Effects of Paddy Cultivation and Microbiota Members on Arsenic Accumulation in Rice Grain"

_foods, 2023, doi:10.3390/foods12112155_

Round 1
Reviewer 1 Report
The manuscript seems good, but ….
Title of Manuscript to accept
Abstract and key words – correct
Introduction
Please mention about phytoremediation As by plants like ...
You can show in book
Antonkiewicz J., Gworek B. 2023. Remediation of contaminated soils and lands. Publisher: Wydawnictwo Naukowe PWN, Warsaw, pp. 2002. DOI: https://doi.org/10.53271/2022.138, ISBN: 978-83-01-22827-9.
Material and method
Were used CRM to chemical analysis ?
Discusion
Please provide volatile forms ….
Conclusions
Maybe Authors can add information about stabilization of As in lands
In the book is information abouit the [problems…..
Antonkiewicz J., Gworek B. 2023. Remediation of contaminated soils and lands. Publisher: Wydawnictwo Naukowe PWN, Warsaw, pp. 2002. DOI: https://doi.org/10.53271/2022.138, ISBN: 978-83-01-22827-9.

Author Response
RESPONSE TO REVIEWER 1
Manuscript ID: foods-2354998
The manuscript seems good, but ….
Title of Manuscript to accept
Abstract and key words – correct
Response: Thank you so much for your valuable comments.
1) Please mention about phytoremediation As by plants like ...
Response: In the introduction section, the sentences “Because, as well as the effect of microbial remediation processes on As accumulation in rice, the phytoremediation feature of the rice has a high effect (Antonkiewicz J., Gworek B. 2023). Therefore, both rice development stages and microbial cycles should be evaluated together for safe rice production and consumption.” were added. Also due to additional reference, the references have been rearranged.
2) For material and method section: Were used CRM to chemical analysis ?.
Response: Certificates of certified reference materials (CRMs) were used for chemical analysis. Because of it, necessary infromations (For chemical analysis, Multi Element ICP QC Standart solution (QCS-27) (27E) was used. It contains 27 elements in 2 to 5% HNO3 + traces HF (QCS-27).) were added in the material and method section.
3) For discussion, Please provide volatile forms …..
Response: AsH3 (arsine), CH3AsH2 (monomethylarsine), (CH3)2AsH (dimethylarsine) and (CH3)3As (trimethylarsine) were added to the discussion section as volatile arsenic forms.
4) For conclusion: Maybe Authors can add information about stabilization of As in lands.
Response: Thank you so much for this book. It is very useful for our manuscript. So, the sentence “At this stage, it can be thought that this situation is due to the ability of paddy to accumulate As or the stabilization properties of As in the soil [27].” was added with the reference for discussion section after the sentences “However, as a result of the highest arsC gene copy numbers obtained in Danakumu sludge samples, it was determined that the highest As accumulation was observed in shoot, rice grain and husk samples obtained from this region.”.
With the informations in this book, conclusion section was revised.
[27]: Antonkiewicz, J.; Gworek, B. Remediation of Contaminated Soils and Lands. Publisher: Wydawnictwo Na-ukowe PWN, Warsaw, 2023, pp. 2002. DOI: https://doi.org/10.53271/2022.138, ISBN: 978-83-01-22827-9.

Reviewer 2 Report
This is a topical study on arsenic levels in rice grain in the context of the potential health concerns related to it. One of the strengths of this study is that it examines various aspects of rice production, including the amount of arsenic in water and soil, changes in genes, abundance and diversity of microbiota, and accumulation of arsenic in different parts of the rice plant. The study also provides specific findings on the highest and lowest values of arsenic accumulation and the abundance of certain microbiota in different water sources. Another strength of this study is its practical implications. My specific recommendation are to include a general and consistent conclusion in the abstract. Table 2 is difficult to read and Figure 1 from A to F uses different types of graphs, you could use the same type for example the one in A, because, in general, the quality of tables and figures is crucial for effectively communicating the study's findings to the reader. The tables and figures should be clear, concise, and well-organized, with appropriate labels, legends, and captions.
Author Response
RESPONSE TO REVIEWER 2
Manuscript ID: foods-2354998
This is a topical study on arsenic levels in rice grain in the context of the potential health concerns related to it. One of the strengths of this study is that it examines various aspects of rice production, including the amount of arsenic in water and soil, changes in genes, abundance and diversity of microbiota, and accumulation of arsenic in different parts of the rice plant. The study also provides specific findings on the highest and lowest values of arsenic accumulation and the abundance of certain microbiota in different water sources. Another strength of this study is its practical implications.
Response: Thank you so much for your valuable comments. Your valuable comments are a source of pride for all our authors.
1) My specific recommendation are to include a general and consistent conclusion in the abstract.
Response: The abstract section has been reviewed and revised from the point of view you specified.
2) Table 2 is difficult to read.
Response: The format of table 2 has been changed for better reading and understanding.
3) Figure 1 from A to F uses different types of graphs, you could use the same type for example the one in A, because, in general, the quality of tables and figures is crucial for effectively communicating the study's findings to the reader.
Response: In order to evaluate the data obtained in the most accurate way, all graphics in Figure 1 have been revised to be in the same format.
3) The tables and figures should be clear, concise, and well-organized, with appropriate labels, legends, and captions.
Response: All tables and figures have been revised to increase intelligibility and evaluability.

Reviewer 3 Report
Please strongly consider improving the discussion. It should be simplified by removing overly elaborate elements of the literature review. Other than that, I have virtually no critical comments.
Author Response
RESPONSE TO REVIEWER 3
Manuscript ID: foods-2354998
Please strongly consider improving the discussion. It should be simplified by removing overly elaborate elements of the literature review. Other than that, I have virtually no critical comments.
Response: Thank you so much for your valuable comments. Considering your important suggestions, the discussion section has been revised and necessary revisions have been made.

Reviewer 4 Report
The manuscript focused on the investigation of how As accumulates in rice grain varying cultivation site and stage. Moreover, this issue has been correlated with microbiome profiles and, tentatively, with specific microbial metabolism putatively linked to As detoxification and reduction of CH4 emission.
So, the paper although it is well explained the research question into the Introduction; the aim and the overall structure need to be rigorously calibrated on the direct meanings of the obtained results.
In my opinion, Materials and methods and Results Paragraphs should be more schematic and immediately comprehensible to the reader.
The study concerns the mapping of As accumulation under various conditions on the one hand, and the characterisation of the telluric microflora on the other. These two approaches must be clearly explained and kept separate in the sub-sections of the Results, showing in fewer words but more schematically the trend of the parameters that have been measured. Similarly, figures and tables should be easy to understand, both in terms of the size of the characters and symbols, and the graphics. in Fig. 1, the pictures are too small with overlapping data and graphs. In Fig. 2 and Fig. 3 I would not use a continuous line as a graph, as the samples are not successive (in time or concentration...). The graphs in Fig. 4, Fig. 5 and Fig. 6 must be self-explanatory.
The Discussion paragraph needs to be rewritten. in 1-2 pages explain what the mechanisms of As accumulation are, what the physiological limits are, and with the support of the literature what the link is between accumulation and microbiological characteristics. Expose the limitations of the functional characterisation of microbial As metabolism. Indicate how the results can be used to reduce the risk of As accumulation.
In the present work, the structure needs more rigour and the discussion reports some conclusions not directly supported by the results obtained
no specific comments
Author Response
RESPONSE TO REVIEWER 4
Manuscript ID: foods-2354998
The manuscript focused on the investigation of how As accumulates in rice grain varying cultivation site and stage. Moreover, this issue has been correlated with microbiome profiles and, tentatively, with specific microbial metabolism putatively linked to As detoxification and reduction of CH4 emission.
Response: Thank you so much for your valuable comments.
- So, the paper although it is well explained the research question into the Introduction; the aim and the overall structure need to be rigorously calibrated on the direct meanings of the obtained results.
Response: Considering your important suggestions, the entire article has been reviewed and necessary revisions have been made. The purpose of the study and the results have been revised so that they can be clearly correlated.
- In my opinion, Materials and methods and Results Paragraphs should be more schematic and immediately comprehensible to the reader.
Response: In order for the study to be better understood by the reader, revisions were made from the content, tables and figures in the material-method and results sections.
- The study concerns the mapping of As accumulation under various conditions on the one hand, and the characterisation of the telluric microflora on the other. These two approaches must be clearly explained and kept separate in the sub-sections of the Results, showing in fewer words but more schematically the trend of the parameters that have been measured. Similarly, figures and tables should be easy to understand, both in terms of the size of the characters and symbols, and the graphics. in Fig. 1, the pictures are too small with overlapping data and graphs. In Fig. 2 and Fig. 3 I would not use a continuous line as a graph, as the samples are not successive (in time or concentration...). The graphs in Fig. 4, Fig. 5 and Fig. 6 must be self-explanatory.
Response: All tables and figures have been revised in terms of format, content and layout to increase clarity. At the same time, some updates were made in the explanations of tables and figures.
- The Discussion paragraph needs to be rewritten. in 1-2 pages explain what the mechanisms of As accumulation are, what the physiological limits are, and with the support of the literature what the link is between accumulation and microbiological characteristics. Expose the limitations of the functional characterisation of microbial As metabolism. Indicate how the results can be used to reduce the risk of As accumulation.
Response: Considering your valuable suggestions, the discussion and conclusion sections have been revised and rearranged.
- In the present work, the structure needs more rigour and the discussion reports some conclusions not directly supported by the results obtained.
Response: The information used for the purpose of interpreting the data obtained as a result of the study was reviewed and necessary revisions were made in line with the suggestions.

Reviewer 5 Report
In the present study, the authors evaluated the amount of arsenic in the water and soil used in the rice development stage, the change in 11 genes of arsC and mcrA by qRT-PCR, and the abundance and diversity of the 12 dominant microbiota. However, the manuscript has the following concerns.
The manuscript is written in very poor English. The whole text needs to be rewritten and English writing should be checked by a native English-speaking expert.
Abstract: It is better to present the results in the form of quantitative in the abstract not only descriptive
Material and methods: I did not find anything about statistical analysis! what was the experimental design? detail of the design must be included. what was the compare means test? What software was used? many things are unclear
Figure 1. None of these comparisons are statistical. Significant differences should be determined statistically and presented at the 5 or 1 percent probability level based on a valid statistical test. Based on the results of mean comparisons, significant differences should be displayed with letters at the top of the columns.
Figure 2. Please add the unit for the vertical axis and the full word of Cp to the chart.

The manuscript is written in very poor English. The whole text needs to be rewritten and English writing should be checked by a native English-speaking expert.
Author Response
RESPONSE TO REVIEWER 5
Manuscript ID: foods-2354998
In the present study, the authors evaluated the amount of arsenic in the water and soil used in the rice development stage, the change in 11 genes of arsC and mcrA by qRT-PCR, and the abundance and diversity of the 12 dominant microbiota. However, the manuscript has the following concerns.
Response: Thank you so much for your valuable comments.
- The manuscript is written in very poor English. The whole text needs to be rewritten and English writing should be checked by a native English-speaking expert.
Response: The whole manuscript was reviewed and again language checked by a native English speaker.
- Abstract: It is better to present the results in the form of quantitative in the abstract not only descriptive
Response: In the abstract section, taking care not to exceed the word limit, quantitative values are also included for the appropriate data along with descriptive expressions.
- Material and methods: I did not find anything about statistical analysis! what was the experimental design? detail of the design must be included. what was the compare means test? What software was used? many things are unclear.
Response: In particular, the analyzes of the data obtained as a result of chemical analysis were carried out using Microsoft Ecxel (Microsoft, Redmond, WA). All graphics in Figure 1 have been revised by adding standard deviation values. In the sections for which the details are not given, the publication from which the method is taken has been referenced in order not to re-enter it.
- Figure 1. None of these comparisons are statistical. Significant differences should be determined statistically and presented at the 5 or 1 percent probability level based on a valid statistical test. Based on the results of mean comparisons, significant differences should be displayed with letters at the top of the columns.
Response: In particular, figure 1 has been revised in terms of format, content, intelligibility and standard deviation data.
- Figure 2. Please add the unit for the vertical axis and the full word of Cp to the chart.
Response: The unit of the vertical axis and the full word and abbreviation of CP in Figure 2 were added where necessary. Also, all tables and figures in the manuscript were revised for good and correct understanding.
- The manuscript is written in very poor English. The whole text needs to be rewritten and English writing should be checked by a native English-speaking expert.
Response: Considering your suggestions, the manuscript was evaluated in terms of language, and the article was redesigned and checked by two independent experts whose native language is English.

Round 2
Reviewer 4 Report
Many of the suggested changes have been made. However, the graphs in Fig. 1 remain unreadable. Also, it would be good to remove the related tables otherwise there is duplication of data and confusion.
The discussion paragraph should be reduced to half and, as already indicated, state: mechanisms of As accumulation; physiological limits of accumulation; and the relationship with soil microbiology. Furthermore, expose the limitations of the functional characterisation of microbial As metabolism.
Please, indicate how the results can be used to reduce the risk of As accumulation. This is not clear.
Major revisions are required.
no comment
Author Response
RESPONSE TO REVIEWER 4
Manuscript ID: foods-2354998
Many of the suggested changes have been made.
Response: Thank you so much for your valuable comments.
- However, the graphs in Fig. 1 remain unreadable. Also, it would be good to remove the related tables otherwise there is duplication of data and confusion.
Response: Considering your evaluations, Figure 1 has been revised again and the table has been drawn. GraphPad Prism 9.0 program was used during the revision. The same revisions have also been made in the graphics in the supplementary file.
- The discussion paragraph should be reduced to half and, as already indicated, state: mechanisms of As accumulation; physiological limits of accumulation; and the relationship with soil microbiology. Furthermore, expose the limitations of the functional characterisation of microbial As metabolism.
Response: Considering your suggestions, a revision was made by adding reference in the discussion and conclusion sections.
- Please, indicate how the results can be used to reduce the risk of As accumulation. This is not clear.
Response: A revision has been made in the discussion and conclusion sections on how the results obtained will be used in reducing the risk of arsenic.

Reviewer 5 Report
The authors have addressed the most of comments and suggestions.
Moderate editing of English language
Author Response
RESPONSE TO REVIEWER 5
Manuscript ID: foods-2354998
The authors have addressed the most of comments and suggestions.
Response: Thank you so much for your valuable comments.
- Moderate editing of English language
Response: The whole manuscript was checked again by another expert whose native language is English and necessary revisions were made.
